# Estrogen profiling in blood and brain: Effects of season and an aggressive interaction in a songbird

Cecilia Jalabert [1,2,3], Megan Q. Liu[2,4,5], Kiran K. Soma [1,2,4,6] *

1 Department of Zoology, University of British Columbia, Vancouver, British Columbia, Canada, 2 Djavad Mowafaghian Centre for Brain Health, University of British Columbia, Vancouver, British Columbia, Canada, 3 Departamento de Neurofisiología Celular y Molecular, Instituto de Investigaciones Biológicas Clemente Estable, Ministerio de Educación y Cultura, Montevideo, Uruguay, 4 Graduate Program in Neuroscience, University of British Columbia, Vancouver, British Columbia, Canada, 5 Integrated Program in Neuroscience, McGill University, Montreal, Quebec, Canada, 6 Department of Psychology, University of British Columbia, Vancouver, British Columbia, Canada

* ksoma@psych.ubc.ca

## Abstract

Neuroestrogens are synthesized in the brain and regulate social behavior and cognition. In the song sparrow (*Melospiza melodia*), 17β-estradiol (17β-$E_2$) promotes aggression, even during the non-breeding season, when circulating 17β-$E_2$ levels are non-detectable. Measuring estrogens is challenging due to their low concentrations and the limited sensitivity of many existing assays. Moreover, estrogens other than 17β-$E_2$ are often overlooked. Here, we developed a liquid chromatography tandem mass spectrometry (LC-MS/MS) assay for the simultaneous measurement of eleven estrogens using derivatization with 1,2-dimethylimidazole-5-sulfonyl-chloride (DMIS) to enhance sensitivity. We included four $^{13}$C-labeled internal standards, which corrected for matrix effects when measuring catecholestrogens and methoxyestrogens. This method is highly specific, sensitive, accurate, and precise, and improves upon our prior protocol, which could measure four estrogens with a single deuterated internal standard. Then, we applied the new method to samples from free-living adult male song sparrows to assess the effects of season and an aggressive encounter on blood and brain estrogens. Subjects were randomly assigned to either a 10-min simulated territorial intrusion (STI; song playback and live decoy) or a control condition (silent speaker and empty cage) during the breeding or non-breeding season, followed by rapid capture and immediate collection of blood and brain tissue. Only estrone and 17β-$E_2$ were present in blood and brain, while the other nine estrogens in the panel were below detection limits. As expected, there was large regional variation in neuroestrogen levels and very low estrogen levels in blood. There was also large seasonal variation, and estrogen levels were lower in the non-breeding season. Despite robust aggression to the STI, estrogen levels did not differ between STI and control subjects in either season. In sum, our novel method enables ultrasensitive

**Data availability statement:** All data files are available from the OSF database, with total open access at https://osf.io/5dzyj/?view_only=d-7c856d6ba7e4d38b9aebf8933beca42.

**Funding:** This study was supported by Canadian Institutes of Health Research in the form of a grant awarded to KKS (169203) and Canada Foundation for Innovation in the form of a grant awarded to KKS (32631). This study was supported by Agencia Nacional de Investigación e Innovación and UBC Zoology Department in the form of a salary for CJ, and Canadian Institutes of Health Research and UBC Faculty of Medicine in the form of a salary for MQL. The specific roles of these authors are articulated in the 'author contributions' section. The funders had no role in study design, data collection and analysis, decision to publish, or preparation of the manuscript.

**Competing interests:** NO authors have competing interests.

measurement of eleven estrogens and will be useful for studies of songbirds and other animals.

## Introduction

Estrogens play critical roles in physiology and behavior. The most widely studied estrogen, 17β-estradiol (17β-$E_2$), is traditionally associated with female reproductive behavior, but also modulates memory, attention, executive functions, and multiple social behaviors in females and males [1–7]. In fact, 17β-$E_2$ administration rapidly increases aggression in male fish, birds, and mammals [8–12]. Furthermore, social interactions modulate estrogen levels, which may in turn, influence future social behavior [13–16].

The aromatase enzyme (CYP19A1), responsible for estrogen synthesis, is expressed in several brain regions that regulate social behavior [17–21], revealing the capability of local estrogen synthesis. Brain-synthesized estrogens (neuroestrogens) can exert rapid effects near the site of production. Neuroestrogens exhibit rapid fluctuations, as brain aromatase activity is modulated by social interactions, such as courtship [22] and competition [23,24]. Rapid changes in neuroestrogen levels occur even in the absence of changes in circulating estrogen levels. In male zebra finches (*Taeniopygia guttata*), 17β-$E_2$ levels in a forebrain region increase rapidly during an interaction with a female, despite stable circulating 17β-$E_2$ levels. This effect is abolished by intracerebral administration of an aromatase inhibitor [25].

Song sparrows (*Melospiza melodia*) are a well-established songbird model for studying the steroid regulation of aggression under natural conditions. In the Pacific Northwest, song sparrows breed from March through August [26]. Males exhibit robust territorial aggression during both the breeding and non-breeding seasons, despite the lack of circulating androgens and estrogens in the non-breeding season [27]. Aromatase is expressed in brain regions implicated in social behavior [28], and increased aromatase activity in the ventromedial telencephalon is associated with elevated aggression in male song sparrows [19]. Additionally, inhibiting estrogen synthesis during the non-breeding season reduces aggressive responses to a simulated territorial intrusion (STI), an effect that is rescued by 17β-$E_2$ replacement [29,30]. Furthermore, 17β-$E_2$ administration to non-breeding males rapidly increases aggression within 20 min [31].

We recently investigated the rapid effects of an aggressive interaction on brain and blood steroid levels in wild song sparrows [27]. During the breeding season, androgens were detectable but not rapidly modulated in blood or brain regions. In the non-breeding season, androgens were non-detectable in blood of control and STI subjects; however, androstenedione and testosterone increased rapidly in specific brain regions following a STI. Circulating estrogens were not detected during the breeding season, although brain estrogens were present. In the non-breeding season, estrogens were non-detectable in both control and STI subjects. STI did not alter estrogen levels in blood or brain during either season. However, in that study, the

assay might not have been sufficiently sensitive to measure very low levels of estrogens in microdissected brain regions. We have very recently developed methods with even greater sensitivity for estrogens.

Liquid chromatography-tandem mass spectrometry (LC-MS/MS) is the gold standard for steroid quantification [32]. Compared to antibody-based assays, LC-MS/MS offers higher specificity and sensitivity and enables the simultaneous quantification of multiple steroids [33–39]. Since estrogens are often present at very low concentrations, assay sensitivity is critical [40,41]. Derivatization can improve ionization efficiency and increase LC-MS/MS assay sensitivity [42–48]. We previously used an estrogen-specific derivatization reagent 1,2-dimethylimidazole-5-sulfonyl-chloride (DMIS) to quantify four estrogens: estrone ($E_1$), 17β-$E_2$, 17α-estradiol (17α-$E_2$), and estriol ($E_3$) in songbird blood, plasma, and brain tissue [46], based on previous studies in humans and mice [45,49]. We also attempted to measure 4-hydroxyestradiol (4OH-$E_2$), 2-methoxyestradiol (2Me-$E_2$), and 4-methoxyestradiol (4Me-$E_2$) but matrix effects were very high for those analytes. In breeding males that were not exposed to an aggressive challenge, $E_1$ and 17β-$E_2$ levels were much higher in the brain than in the circulation, and local levels of $E_1$ and 17β-$E_2$ showed regional variation that aligned with aromatase expression. In non-breeding males that were not exposed to an aggressive challenge, we detected 17β-$E_2$ only in the caudomedial nidopallium. This previous study did not include STI subjects. Moreover, this study used only one internal standard (IS), deuterated 17β-$E_2$. It would be useful to include additional ISs, additional estrogens, and subjects exposed to an STI.

Here, we improved the estrogen quantification method by using four [13]C-labeled ISs, and we added four estrogens: 16α-hydroxyestrone (16α-OH-$E_1$), 2-hydroxyestrone (2OH-$E_1$), 2-methoxyestrone (2Me-$E_1$), and 4-methoxyestrone (4Me-$E_1$). This resulted in a panel of 11 estrogens. Then, using our improved method, we measured estrogens in blood and 10 microdissected brain regions of free-living adult male song sparrows collected during the breeding and non-breeding seasons, with and without an STI.

## Materials and methods

### Field procedures

The study was conducted on free-living adult male song sparrows during both the breeding season (May 8–24, 2019) and non-breeding season (Nov 2-Dec 2, 2019). All breeding males had an enlarged cloacal protuberance and large testes. All non-breeding males had a small cloacal protuberance and fully regressed testes, and they also were post-molt. Field sites were located near Vancouver, British Columbia, Canada. A total of 45 individuals were used: 5 for assay development and validation and 40 for method application (see below). All subjects were captured in the same manner, using a mist net and conspecific song playback. Immediately after capture, the subjects were deeply anesthetized with isoflurane to minimize suffering and euthanized by rapid decapitation within 3 min of capture to minimize handling effects. The brain was quickly collected and snap-frozen on powdered dry ice. Trunk blood was collected in heparinized microhematocrit tubes (Fisher Scientific) and kept on wet ice until transported to the laboratory (within 5 h). In the laboratory, both blood and brain samples were stored at –70°C.

All animal procedures complied with the Canadian Council on Animal Care, and protocols were approved by the Canadian Wildlife Service (Permit #SC-BC-2019–0012) and the University of British Columbia Animal Care Committee (Protocol #A19-0074).

### Reagents

High Performance Liquid Chromatography (HPLC)-grade acetone, acetonitrile, hexane, and methanol were from Fisher Chemical. The derivatization reagent, 1,2-dimethylimidazole-5-sulfonyl-chloride (DMIS), was purchased from Apollo Scientific (Stockport United Kingdom, Lot number: AS478881, CAS number: 849351-92-4) [49]. Powdered DMIS was stored at 4°C under nitrogen gas and protected from light and moisture [46]. Once opened, powdered DMIS was aliquoted and stored at 4°C for up to 12 months. On the day of derivatization, acetone was added to individual aliquots of DMIS to prepare a fresh solution of 1 mg/mL. Sodium bicarbonate buffer (50mM, pH 10.5) was prepared in Milli-Q water.

Here, we studied a panel of 11 estrogens: $E_1$, 17β-$E_2$, 17α-$E_2$, $E_3$, 4OH-$E_2$, 2Me-$E_2$, 4Me-$E_2$, 16α-OH-$E_1$, 2OH-$E_1$, 2Me-$E_1$, and 4Me-$E_1$. The last 4 analytes were newly added to our previous estrogen panel [46]. All stock solutions were prepared in HPLC-grade methanol. Certified reference standards for $E_1$, 17β-$E_2$, and $E_3$ were obtained from Cerilliant. 17α-$E_2$, 2Me-$E_2$, 4Me-$E_2$, 4OH-$E_2$, 16α-OH-$E_1$, 2OH-$E_1$, 2Me-$E_1$, and 4Me-$E_1$ were obtained from Steraloids. Since 17α-$E_2$ has the same transitions as 17β-$E_2$, with retention times differing by only 0.14 min (resulting in overlapping peaks), 17α-$E_2$ was included in a separate calibration curve. Calibration curves were prepared in 50% methanol and consisted of 10 points ranging from 0.01 to 20 pg per tube for all analytes, except for 2OH-$E_1$ and 4OH-$E_2$, which ranged from 0.1 to 200 pg per tube. The lower limit of quantification (LLOQ) was determined as the lowest standard on the calibration curve for which the analyte peak had a signal-to-noise ratio >10.

The present study also expanded and improved upon our previous use of IS. Previously, we used only deuterated 17β-$E_2$ [46]. Here we employed four $^{13}C$-labeled ISs, which are more stable and more closely match the retention times of their analytes than deuterated ISs. The ISs used here were 2,3,4-$^{13}C_3$-estriol ($^{13}C_3$-$E_3$); 13,14,15,16,17,18-$^{13}C_6$-17β-estradiol ($^{13}C_6$-$E_2$); 13,14,15,16,17,18-$^{13}C_6$-2-methoxyestradiol ($^{13}C_6$-2Me-$E_2$); and 13,14,15,16,17,18-$^{13}C_6$-2-hydroxyestradiol ($^{13}C_6$-2OH-$E_2$), all purchased from Cambridge Isotope Laboratories. Each IS was used for analytes with a similar structure and retention time; $^{13}C_3$-$E_3$ for $E_3$ and 16α-OH-$E_1$; $^{13}C_6$-$E_2$ for $E_1$, 17β-$E_2$, and 17α-$E_2$; $^{13}C_6$-2Me-$E_2$ for 2Me-$E_1$, 4Me-$E_1$, 2Me-$E_2$, and 4Me-$E_2$; and $^{13}C_6$-2OH-$E_2$ for 2OH-$E_1$ and 4OH-$E_2$. This grouping ensured optimal correction for matrix effects and variability in recovery across chemically related analytes. Each IS was prepared to a final working solution of 40 pg/mL in 50% methanol.

## Estrogen extraction and derivatization

Estrogens were extracted from brain tissue or blood samples (20 μL) using liquid−liquid extraction as before [27,46,50]. Briefly, 1 mL of acetonitrile was added to each sample, and 50 μL of IS (i.e., 2 pg of each IS) was added to all samples except "double blanks." Samples were homogenized using a bead mill homogenizer (Omni International Inc., Kennesaw, GA, USA) at 4 m/s for 30 sec. Following homogenization, samples were centrifuged at 16,100 g for 5 min, and 1 mL of the supernatant was transferred to a pre-cleaned borosilicate glass culture tube (12 x 75 mm). Then, 500 μl of hexane was added, and tubes were vortexed and centrifuged at 3200 g for 2 min. The hexane layer was removed and discarded, and extracts were dried at 60°C for 45 min in a vacuum centrifuge. Calibration curves, quality control (QC) samples, blanks, and double blanks were prepared alongside biological samples.

Derivatization was performed as before [46], based on earlier studies [45,49]. Dried extracts were placed in an ice bath and reconstituted with 30 μL of sodium bicarbonate buffer (50 mM, pH 10.5), briefly vortexed, and 20 μL of 1 mg/mL DMIS in acetone was added. The samples were vortexed, centrifuged at 3200 g for 1 min, and then transferred to glass LC-MS vial inserts in LC-MS vials (Agilent, Santa Clara, CA, USA). The vials were capped and incubated at 60°C for 15 min, followed by a cooling period of 15 min at 4°C. Finally, the samples were centrifuged at 3200 g for 1 min and then stored at −20°C for a maximum of 24 h before steroid analysis.

## Estrogen analysis by LC-MS/MS

Estrogens were quantified as previously described [46]. Samples were placed in a refrigerated (15°C) autoinjector in a Nexera X2 UHPLC (Shimadzu Corp., Kyoto, Japan) and 35 μL per sample were passed through a KrudKatcher ULTRA HPLC In-Line Filter (Phenomenex, Torrance, CA) and then an Agilent 120 HPH C18 guard column (2.1 mm). Then, estrogens were separated on an Agilent 120 HPH C18 column (2.1 x 50 mm; 2.7 μm; at 40°C) using 0.1 mM ammonium fluoride in MilliQ water as mobile phase A (MPA) and methanol as mobile phase B (MPB). The flow rate was 0.4 mL/min. During loading, MPB was at 10% for 1.6 min, and from 1.6 to 4 min the gradient profile was at 42% MPB, which was ramped up to 60% MPB until 9.4 min. From 9.4 to 11.9 min the gradient was ramped from 60% to 98% MPB. Finally, a column wash was performed from 11.9 to 13.4 min at 98% MPB. The MPB was then returned to starting conditions of

10% MPB for 1.5 min. Total run time was 14.9 min. The needle was rinsed externally with 100% isopropanol before and after each sample injection.

We used two multiple reaction monitoring (MRM) transitions for each analyte and one MRM transition for each IS (Table 1). Steroid concentrations were acquired on a Sciex 6500 Qtrap triple quadrupole tandem mass spectrometer (Sciex LLC, Framingham, MA) in positive electrospray ionization mode for derivatized estrogens and negative electrospray ionization mode for underivatized estrogens. We monitored underivatized estrogens and confirmed that the derivatization reaction was complete. No analyte peaks were detected in the blanks and double blanks.

## Assay accuracy and precision

Accuracy and precision were assessed using in-house prepared QCs with known amounts of standard in neat solution at 2 concentrations. The low concentration QC had 0.5 pg, and the high concentration QC had 2 pg of each estrogen, except $2OH-E_1$ and $4OH-E_2$ which had 5 and 20 pg respectively. For $17\alpha-E_2$ only the low 0.5 pg QC was used. Biological samples were analyzed across six different assays, with three replicates of each concentration in each assay (n = 36 QCs total). Accuracy was determined by comparing the measured values to the known values. Precision was determined by calculating the coefficient of variation of QC measurements within the same run (intra-assay variation) and across different runs (inter-assay variation).

## Matrix effects

Matrix effects were assessed in blood samples (20 μL) and brain samples (1.5 mg) from a total of n = 5 animals. First, we evaluated estrogen recovery from biological matrices. To this end, pooled blood and pooled brain samples were each divided into two aliquots: one (unspiked) was analyzed to determine endogenous estrogen concentrations, and the other was spiked with a known amount of standard prior to extraction to determine the effects of each biological matrix on steroid recovery. A third tube, containing the same amount of standard in a neat solution (i.e., 50% methanol solvent only), served as the reference. Each tube was assessed in 5 replicates. Recovery (%) was calculated by comparing the signal

Table 1. Scheduled multiple reaction monitoring for derivatized estrogens.

| Analyte | Retention time (min) | Quantifier m/z | Qualifier m/z |
| --- | --- | --- | --- |
| $E_1$ | 10.46 | 429→365 | 429→96 |
| $17\beta-E_2$ | 10.74 | 431→367 | 431→96 |
| $17\alpha-E_2$ | 10.88 | 431→367 | 431→96 |
| $^{13}C_6-E_2$ | 10.74 | 437→373 | – |
| $16\alpha-OHE_1$ | 8.25 | 445→381 | 445→96 |
| $E_3$ | 7.97 | 447→383 | 447→96 |
| $^{13}C_3-E_3$ | 7.97 | 450→386 | – |
| $2Me-E_1$ | 10.74 | 459→300 | 459→299 |
| $4Me-E_1$ | 10.42 | 459→299 | 459→97 |
| $2Me-E_2$ | 11.04 | 461→302 | 461→283 |
| $4Me-E_2$ | 10.57 | 461→283 | 461→161 |
| $^{13}C_6-2Me-E_2$ | 11.04 | 467→308 | – |
| $2OH-E_1$ | 10.23 | 603→380 | 603→96 |
| $4OH-E_2$ | 10.62 | 605→382 | 605→96 |
| $^{13}C_6-2OH-E_2$ | 10.62 | 611→388 | – |

For all $^{13}C$-labeled ISs, only one transition was monitored.

from the spiked biological sample (after subtracting endogenous levels from the unspiked sample) to the signal obtained from the spiked neat solution.

Second, since we added the same amount of IS in all samples, we also compared the IS peak areas in blood and brain samples to those in neat solution. Differences in IS peak area of less than 20% were considered acceptable. A decrease in IS peak area of more than 20% indicated ion suppression, while an increase of more than 20% indicated ion enhancement.

### Behavior and tissue collection for method application

We quantified 11 estrogens in wild male song sparrows during both the breeding and non-breeding season. In each season, subjects were randomly assigned to one of two treatment groups: simulated territorial intrusion (STI) or control (CON) (n = 10 subjects per group per season). Subjects in the STI group were exposed to a conspecific song playback from a speaker and a live caged decoy placed within their territory for 10 min. In contrast, subjects in the CON group were exposed to an empty cage and a silent speaker [27]. During the CON and STI conditions, aggressive behaviors were documented in real time in field notebooks [51,52], including song latency, flight latency, number of songs, number of flights, and time spent within 5 m of the decoy. Because song sparrows defend well-defined and non-overlapping territories, only the focal individual responded strongly to the STI, ensuring that behavioral observations and subsequent sampling were restricted to a single subject.

Then, a mist net (set up in advance) was quickly unfurled and subjects were captured as described in the field procedures subsection. The playback used for capture was limited to a maximum of 5 min to avoid potential effects on steroid levels and did not differ significantly across groups (breeding CON: 2.2±0.6 min, breeding STI: 1.4±0.6 min, non-breeding CON: 2.0±0.4 min, non-breeding STI: 1.5±0.4 min, $F_{3,36}$ = 1.27, p = 0.30). Immediately after capture, the subjects were euthanized within 3 min of capture, and handling duration was similar across groups (breeding CON: 2.5±0.1 min, breeding STI: 2.5±0.2 min, non-breeding CON: 2.1±0.1 min, non-breeding STI: 2.6±0.1 min, $F_{3,36}$ = 2.61, p = 0.066).

### Brain dissection

The Palkovits punch technique [53] was used to microdissect 9 brain regions that regulate social behavior (for details, see [27,46,50]). These regions include the caudal portion of the preoptic area (POA), anterior hypothalamus (AH), lateral septum (LS), bed nucleus of the stria terminalis (BNST), ventromedial hypothalamus (VMH), ventral tegmental area (VTA), central grey (CG), caudomedial nidopallium (NCM), and nucleus taeniae of the amygdala (TnA) (homolog of the mammalian medial amygdala). We also included the cerebellum (Cb), which has low aromatase expression and is not in the social behavior network. Brains were sectioned coronally at 300 µm using a MicroHM525 cryostat (Thermo Fisher Scientific Inc., Waltham, MA), and bilateral punches were collected. The same punch size (1 mm diameter) was used for all regions. Depending on the size of the region, either 4 or 6 punches were collected, yielding 0.98 or 1.47 mg of tissue, respectively. Punches were expelled into cold 2-mL polypropylene tubes (Sarstedt AG & Co, Numbrecht Germany), each containing five zirconium ceramic oxide beads (1.4-mm diameter) and stored at –70°C.

### Statistical analysis

A value was considered below the LLOQ if it fell below the lowest standard on the calibration curve. When 20% or more of the samples in a group (blood or brain region) were above the LLOQ, then the values below the LLOQ were imputed as before [27,46,50,54,55]. If less than 20% of the samples in a group were above the LLOQ, then imputations were not performed.

Statistical analyses were conducted using GraphPad Prism version 10.3.1 (GraphPad Software). When necessary, data were log-transformed prior to analysis. Seasonal differences in behavior were analyzed by unpaired t-tests. To examine the effects of STI in blood, we used unpaired t-tests. To examine the effects of STI in the brain, we used repeated

mixed-measures two-way ANOVA with a between-subjects factor (STI) and a within-subjects factor (region). The correlations between $E_1$ and $17\beta\text{-}E_2$ levels in the brain were examined using Spearman's rho correlations. The significance criterion was set at $p \leq 0.05$. Graphs show the mean±SEM and are presented with non-transformed data.

## Results

### Assay development and validation

An assay for quantifying 11 estrogens was successfully developed using DMIS derivatization and LC-MS/MS. First, we achieved high specificity by identifying at least one MRM transition that is specific to each estrogen, and through the optimization of the liquid chromatography, which allowed the separation of isomers that shared MRM transitions (Table 1). Second, assay sensitivity was greatly enhanced for all estrogens after DMIS derivatization, as evidenced by the improved LLOQs (Table 2) and the linearity across the range (Fig 1). Third, assay accuracy, as indicated by low and high QCs, generally fell within an acceptable range (100±20%), with the exception of $4\text{OH-}E_2$ which was below 80% (Table 2). Fourth, assay precision, as indicated by intra- and inter-assay coefficients of variation, was considered acceptable when lower than 20%. The analytes $E_1$, $17\beta\text{-}E_2$, $E_3$, $2\text{Me-}E_1$, and $16\alpha\text{-OH-}E_1$ showed acceptable precision at both low and high QC concentrations (Table 2), and for the rest of the analytes, variability was higher than desired for either low or high concentration QCs. Lastly, no peaks were detected in all blanks and double blanks for all analytes.

Four $^{13}\text{C}$-labeled estrogens were used as ISs. In general, $^{13}\text{C}$-labeled ISs are preferred over deuterated ISs because $^{13}\text{C}$-labeled ISs are more stable and their retention times more closely match those of the analytes (Fig 2). One $^{13}\text{C}$-labeled estrogen was used for each group of estrogens. $^{13}\text{C}_3\text{-}E_3$ was used for $E_3$ and $16\alpha\text{-OH-}E_1$; $^{13}\text{C}_6\text{-}E_2$ was used for $E_1$, $17\beta\text{-}E_2$, and $17\alpha\text{-}E_2$; $^{13}\text{C}_6\text{-}2\text{Me-}E_2$ was used for $2\text{Me-}E_1$, $4\text{Me-}E_1$, $2\text{Me-}E_2$, and $4\text{Me-}E_2$; and $^{13}\text{C}_6\text{-}2\text{OH-}E_2$ was used for $2\text{OH-}E_1$ and $4\text{OH-}E_2$.

Matrix effects were assessed in blood (20 µL) and brain samples (1.5 mg). A total of n = 5 subjects were used for method development. To assess recovery, we compared estrogen levels in samples spiked with known amounts of standard (after subtracting the endogenous concentrations of unspiked tissue samples) to those of standards spiked in a neat solution. Recovery percentages within the range of 100±20% are considered acceptable. Recovery values for the 11 analytes are presented in Table 3. For $E_1$, $17\beta\text{-}E_2$, $E_3$, and $16\alpha\text{-OHE}_1$ recovery values were within the acceptable range. The

**Table 2. Assay lower limits of quantification (LLOQ), accuracy, and precision.**

| Analyte | LLOQ (pg) | Low concentration Quality Control | | | High concentration Quality Control | | |
|---|---|---|---|---|---|---|---|
| | | Intra-assay CV% | Inter-assay CV% | Accuracy % | Intra-assay CV% | Inter-assay CV% | Accuracy % |
| $E_1$ | 0.02 | 9.1 | 16.1 | 84.9 | 7.3 | 12.4 | 86.3 |
| $17\beta\text{-}E_2$ | 0.05 | 3.9 | 6.0 | 92.8 | 2.8 | 3.7 | 86.1 |
| $17\alpha\text{-}E_2$ | 0.1 | 13.6 | 22.2 | 90.1 | – | – | – |
| $E_3$ | 0.02 | 2.0 | 3.0 | 87.5 | 1.9 | 2.5 | 86.6 |
| $2\text{OH-}E_1$ | 2 | 15.9 | 28.7 | 89.7 | 10.0 | 24.7 | 106.7 |
| $2\text{Me-}E_1$ | 0.2 | 13.3 | 20.6 | 97.5 | 13.4 | 15.5 | 94.5 |
| $4\text{Me-}E_1$ | 1 | – | – | 95.2 | 29.8 | 37.0 | 92.5 |
| $16\alpha\text{-OH-}E_1$ | 0.05 | 3.7 | 7.8 | 85.7 | 3.3 | 5.5 | 84.2 |
| $4\text{OH-}E_2$ | 2 | 14.8 | 27.2 | 76.1 | 23.2 | 28.5 | 105.8 |
| $2\text{Me-}E_2$ | 0.2 | 17.4 | 24.8 | 96.7 | 11.8 | 18.4 | 91.3 |
| $4\text{Me-}E_2$ | 0.5 | 19.0 | 25.2 | 101.3 | 16.8 | 19.1 | 96.2 |

The lower limit of quantification (LLOQ) was defined as the lowest standard on the calibration curve with a signal-to-noise ratio greater than 10. The low quality control (QC) was 0.5 pg for each analyte, except for $2\text{OH-}E_1$ and $4\text{OH-}E_2$ (5 pg). The high QC was 2 pg, except for $2\text{OH-}E_1$ and $4\text{OH-}E_2$ (50 pg). For $4\text{Me-}E_1$ the low QC was below its LLOQ. For $17\alpha\text{-}E_2$ only the low QC was used.

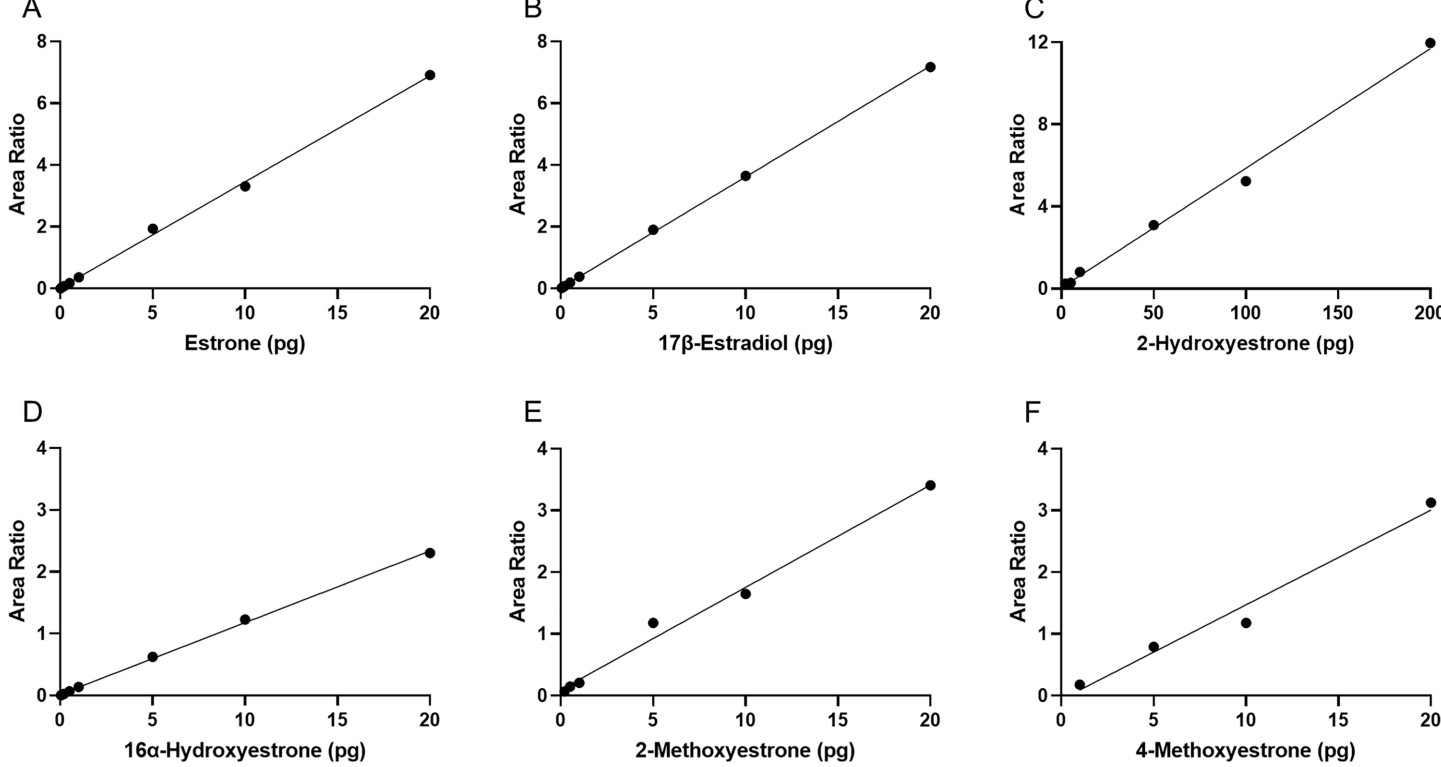

**Fig 1. Calibration curves for 6 derivatized estrogen analytes measured by liquid chromatography-tandem mass spectrometry (LC-MS/MS).** (A) estrone, (B) 17β-estradiol, (C) 2-hydroxyestrone, (D) 16α-hydroxyestrone, (E) 2-methoxyestrone, and (F) 4-methoxyestrone. The calibration curve for 2-hydroxyestrone is one order of magnitude higher than the other analytes, because of lower sensitivity for this analyte. Area ratio is calculated by dividing an analyte peak area by the appropriate internal standard (IS) peak area in the same sample.

catecholestrogens, 2OH-$E_1$ and 4OH-$E_2$, showed acceptable recoveries in brain but lower recoveries in blood. Methoxyestrogens showed variable performance: 4Me-$E_2$ had good recovery; 4Me-$E_1$ showed low recovery in brain; 2Me-$E_1$ and 2Me-$E_2$ showed affected recovery in both brain and blood.

Matrix effects were also assessed by comparing IS peak areas in blood and brain samples to those in neat solution (Table 4). Tissue IS peak areas were considered acceptable in a range of $100 \pm 20\%$. A decrease of more than 20% indicates ion suppression, while an increase of more than 20% indicates ion enhancement. The $^{13}C_6$-$E_2$ IS showed comparable peak areas in both blood and brain relative to the neat solution, suggesting minimal matrix effects. The $^{13}C_3$-$E_3$ IS exhibited potential ion suppression in blood but remained within the acceptable range in brain tissue. In contrast, $^{13}C_6$-2OH-$E_2$ and $^{13}C_6$-2Me-$E_2$ showed higher IS peak areas in both biological matrices, indicating possible ion enhancement. Nonetheless, catecholestrogens and methoxyestrogens analytes demonstrated better recovery performance than their respective IS (Table 3), indicating that the ISs partially corrected for matrix effects.

## Method application in songbird samples

As expected, male song sparrows responded aggressively to the STI during both the breeding and non-breeding seasons, and no significant seasonal differences were found in any of the behaviors assessed (Table 5). These data indicate that the sparrows respond with similar levels of aggression to an intruder during these two seasons, as in previous studies [56,57].

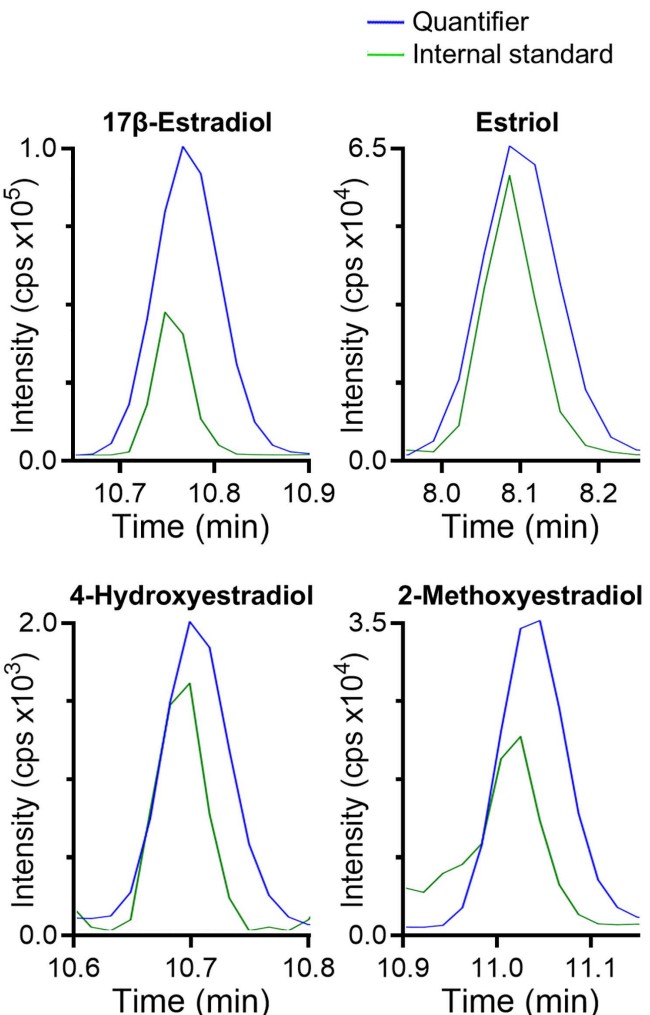

**Fig 2. Representative chromatograms for 4 derivatized estrogen analytes measured by liquid chromatography-tandem mass spectrometry (LC-MS/MS).** The samples were in neat solution for (A) 17β-estradiol (17β-$E_2$), (B) estriol ($E_3$), (C) 4-hydroxyestradiol (4OH-$E_2$), and (D) 2-methoxyestradiol (2Me-$E_2$). The quantifier transitions of the reference standards are shown in blue (10 pg of each) and the $^{13}C$-labeled internal standard transitions are shown in green (2 pg of each). Intensity is measured in counts per second (cps).

**Table 3. Assay recovery %.**

|  | Blood (20 μL) | Brain (1.5 mg) |
|---|---|---|
| $E_1$ | 99 | 106 |
| 17β-$E_2$ | 91 | 105 |
| $E_3$ | 105 | 105 |
| 2OH-$E_1$ | 48 | 86 |
| 2Me-$E_1$ | 211 | 126 |
| 4Me-$E_1$ | 88 | 76 |
| 16α-OH-$E_1$ | 118 | 104 |
| 4OH-$E_2$ | 25 | 113 |
| 2Me-$E_2$ | 74 | 130 |
| 4Me-$E_2$ | 84 | 82 |

**Table 4. Internal standard peak area in each matrix relative to neat solution (%).**

|  | Blood (20 μL) | Brain (1.5 mg) |
|---|---|---|
| $^{13}C_6$-$E_2$ | 87 | 94 |
| $^{13}C_3$-$E_3$ | 62 | 97 |
| $^{13}C_6$-2OH-$E_2$ | 812 | 422 |
| $^{13}C_6$-2Me-$E_2$ | 260 | 188 |

**Table 5. Behavioral responses to a Simulated Territorial Intrusion (STI) in wild song sparrows.**

|  | Breeding Season | Non-breeding Season | t | p |
|---|---|---|---|---|
| Song latency (s) | 125.0±49.2 | 132.1±58.8 | 0.11 | 0.91 |
| Flight latency (s) | 72.2±26.3 | 39.0±10.5 | 0.02 | 0.98 |
| Time in 5 meters (s) | 391.5±56.5 | 486.8±45.5 | 1.22 | 0.24 |
| Number of songs | 30.4±5.5 | 27.5±6.2 | 0.34 | 0.73 |
| Number of flights | 25.4±3.0 | 34.2±3.3 | 1.8 | 0.09 |

STI duration was 10 min. Data are mean±SEM. The p values are from unpaired t-tests. n=10 per group.

We applied the validated method to examine a panel of 11 estrogens in the blood and 10 microdissected brain regions of CON and STI subjects in the two seasons (n=10 per treatment per season). This generated 121 data points per subject, and thus 4,840 data points across 40 subjects total.

In the breeding season, only $E_1$ and 17β-$E_2$ were quantifiable in blood and brain samples, in both the CON and STI groups (Fig 3). For $E_1$, STI had no significant effect on blood levels (t=0.76, p=0.46). In the brain, there was a significant main effect of region on $E_1$ levels (F(9,162) = 39.05; p<0.0001) but no significant main effect of STI (F(1,18) = 0.25; p=0.63) and no significant region×STI interaction (F(9,162) = 0.76; p=0.65). Similarly, for 17β-$E_2$, levels in blood were not significantly different between CON and STI groups, although there was a trend for STI to increase 17β-$E_2$ levels (t=2.01, p=0.06). In the brain, 17β-$E_2$ levels showed a significant main effect of region (F(9,162) = 166.5; p<0.0001) but no significant main effect of STI (F(1,18) = 0.0014; p=0.97) and no significant region×STI interaction (F(9,162) = 0.76; p=0.65).

In the non-breeding season, neither $E_1$ nor 17β-$E_2$ were quantifiable in any samples from both groups.

The other nine estrogens in the panel were below the LLOQs in blood and brain in both groups and both seasons.

Next, in samples from the breeding season, we assessed the relationships between 17β-$E_2$ and its precursor $E_1$ in each brain area. As expected, there were significant positive correlations between $E_1$ and 17β-$E_2$ in both groups combined (Fig 4) in all brain areas, except the VTA where no correlation was observed, probably due to its low estrogen levels.

## Discussion

We developed a quantification method for 11 estrogens, adding 4 new analytes and 4 new $^{13}C$-labeled internal standards to our previous method [46]. Overall, assay performance was satisfactory for $E_1$, 17α-$E_2$, 17β-$E_2$, $E_3$, 4Me-$E_1$, 16α-OH-$E_1$, and 4Me-$E_2$ in blood and brain. For the methoxyestrogens, 2Me-$E_1$ and 2Me-$E_2$, performance was hindered in both matrices. Assay performance for catecholestrogens was acceptable in brain tissue, but matrix effects were observed for these analytes in blood. The improvements presented here are, in part, the result of the four $^{13}C$-labeled IS that we used. We then applied the method to wild song sparrows and examined the effects of tissue, season, and an aggressive encounter. We detected only $E_1$ and 17β-$E_2$. The tissue differences were robust, and $E_1$ and 17β-$E_2$ levels were higher in brain regions that express aromatase. The seasonal effect was also robust, and estrogens were higher during the breeding season than the non-breeding season. Male song sparrows showed intense aggression during both seasons, yet the aggressive encounter did not acutely affect the concentration of $E_1$ and 17β-$E_2$ in brain or blood.

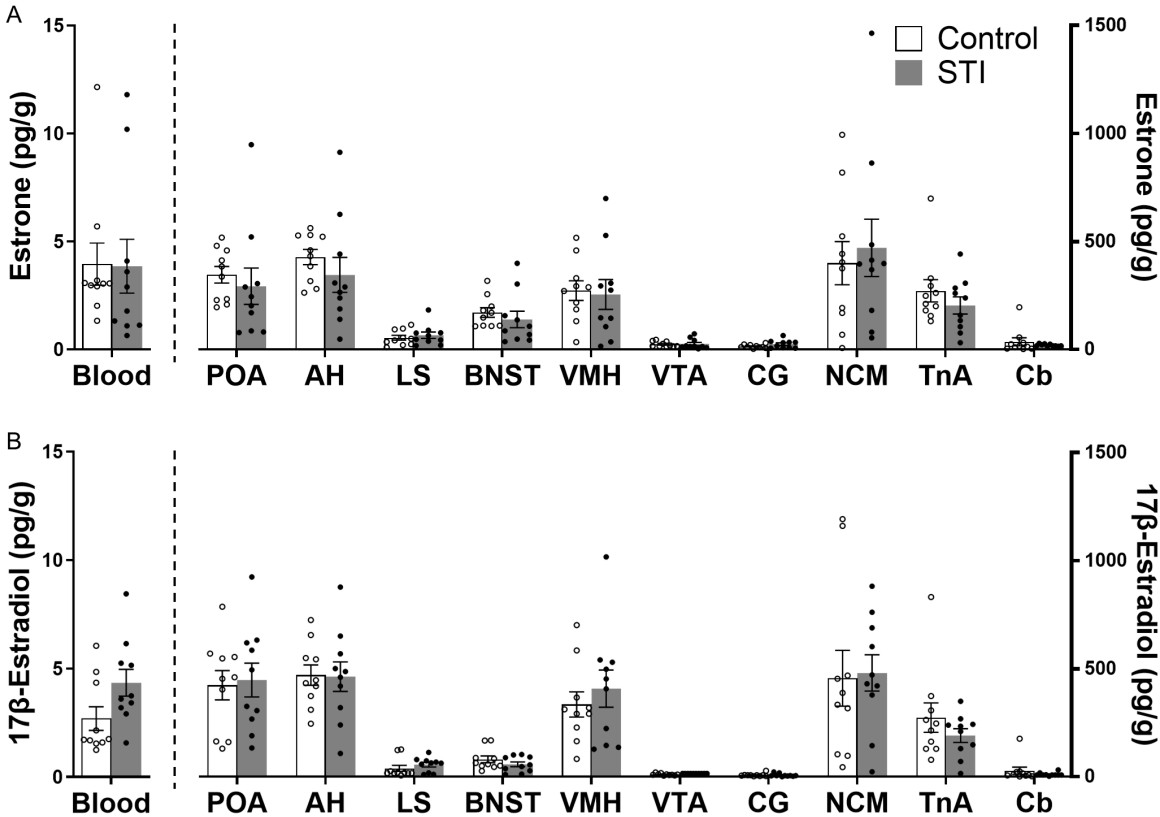

**Fig 3. Lack of effect of a 10-min STI on circulating and brain estrogen levels of wild adult male song sparrows during the breeding season.** The bar graphs represent concentrations of **(A)** $E_1$ and (B) 17β-$E_2$. Values are expressed as the mean ± SEM. n = 10 per group.

## Method development and validation

Estrogen measurement is particularly challenging due in part to low abundance of estrogens in biological samples, but mass spectrometry assay sensitivity can be improved through derivatization [39,40,42,58], as seen here. We separated analytes by ultra-high-performance liquid chromatography (UHPLC), and each analyte exhibited a unique retention time, supporting both analyte identification and method specificity. Here, we used electrospray ionization (ESI). While atmospheric pressure photoionization is also used with DMIS [49], ESI is more widely employed. The DMIS derivatization reagent produced analyte-specific fragmentation patterns, further contributing to assay specificity. This contrasts with dansyl chloride, the most commonly used reagent for 17β-$E_2$ derivatization, which generates a product ion derived from the dansyl moiety itself, and thus is not analyte-specific [59,60]. Our approach therefore represents a methodological improvement by providing enhanced specificity for estrogen detection.

Importantly, most studies of estrogens exclusively measure 17β-$E_2$, and the levels of other estrogens are under-investigated. Both $E_1$ and 17β-$E_2$ undergo hydroxylation at the C2 and C4 positions, generating bioactive catecholestrogens [61,62]. Although relatively labile [63], these compounds have diverse reproductive [64,65], neuromodulatory [66,67], and anti- or pro-carcinogenic effects [68]. These catecholestrogens are converted into methoxyestrogens, which are subsequently conjugated to glucuronide or sulfate groups for excretion [62,69,70]. In this study, we expanded our previous method [46] by incorporating 4 additional estrogens (16α-OH-$E_1$, 2OH-$E_1$, 2Me-$E_1$, and 4Me-$E_1$) to our original panel of $E_1$, 17β-$E_2$, 17α-$E_2$, $E_3$, 4OH-$E_2$, 2Me-$E_2$, and 4Me-$E_2$. Accuracy was acceptable for all analytes, even at the lower

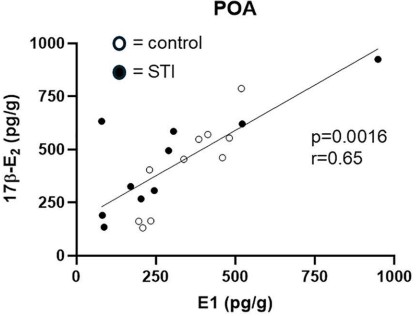

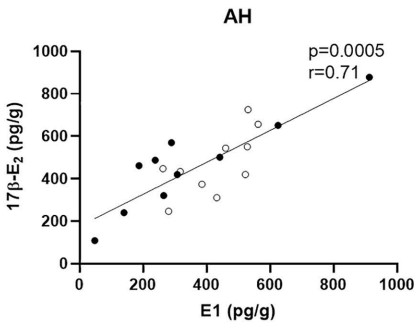

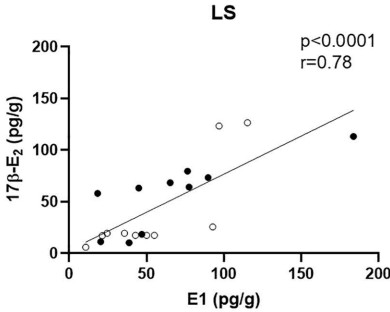

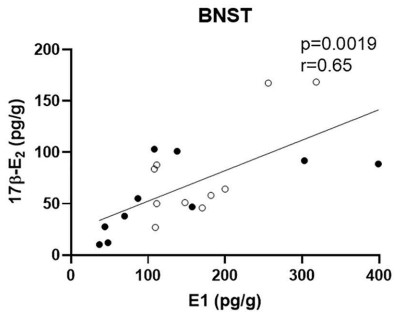

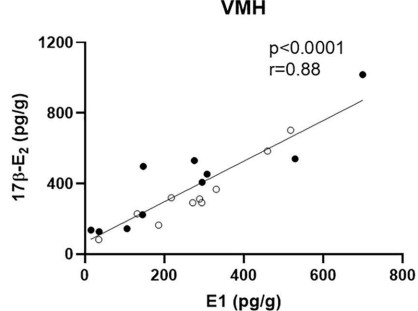

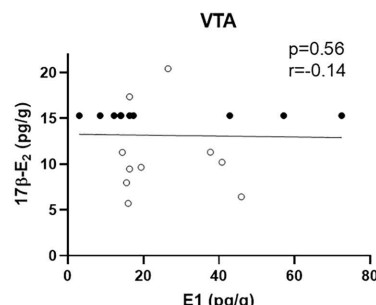

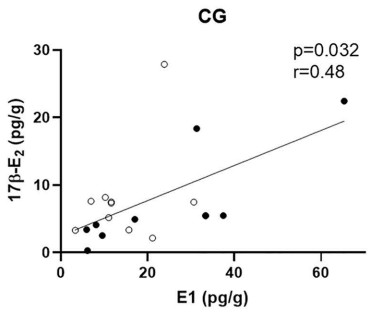

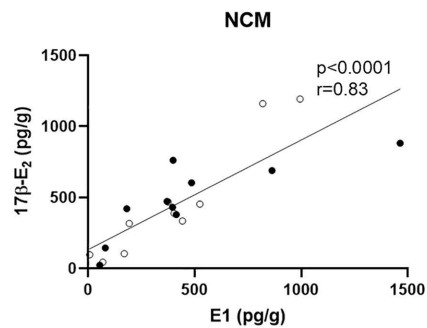

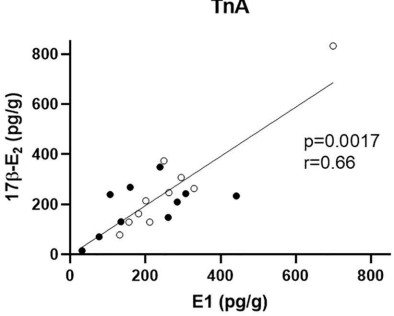

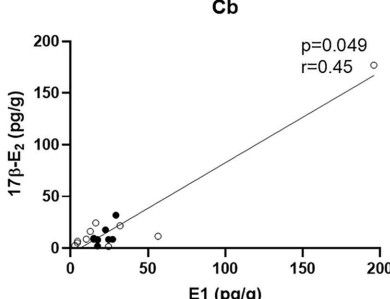

**Fig 4. Correlations between estrogens in each brain area in wild adult male song sparrows during the breeding season.** Levels of $E_1$ and $17\beta$-$E_2$ were significantly positively correlated in all brain regions except the VTA. Data were analyzed using Spearman's rho. Subjects from CON and STI groups are represented with empty and filled circles, respectively. n = 10 per group.

concentrations. While greater variability was observed for the catechol- and methoxy- estrogens, assay precision was acceptable overall. The assay demonstrated excellent sensitivity, as low as 0.02 pg per sample, making this method well-suited for quantifying neuroestrogens in microdissected brain regions, ranging from 0.01 to 2.5 pg per sample [46,50,71], and thus below the LLOQs of most assays.

In the present study we further improved our estrogen quantification method by incorporating four $^{13}$C-labeled ISs. While most studies use deuterated ISs, which are more widely available and affordable, these compounds are more susceptible to hydrogen-deuterium exchange during sample processing, potentially compromising assay accuracy [32]. Additionally, deuterated ISs often exhibit slightly different retention times than their corresponding analytes. In contrast, $^{13}$C-labeled IS more closely match the retention times of their analytes, resulting in improved co-elution and enhanced assay performance [40]. Moreover, in our previous study, we used a single deuterated IS (d4-$17\beta$-$E_2$) for all seven estrogen analytes [46]. In contrast, in the present study, we used four stable-isotope labeled IS, with one labeled estrogen per chemical group. This refinement improved correction for matrix effects, particularly for catechol- and methoxy- estrogens. For example, recovery of 4OH-$E_2$ in our previous study was 613% in brain, but 113% here (similar results are observed for 2Me-$E_2$ and 4Me-$E_2$). Thus, although some matrix effects were observed here, we notably improved analyte recovery compared to our previous method, in which catechol- and methoxy- estrogens were strongly affected by matrix effects. Furthermore, the assay showed better performance in brain tissue, highlighting its suitability for neurosteroid analysis.

Further evidence supports our assay performance. Estrogen levels detected in the present study were similar to those found in previous studies in this species, either with [46] or without [27,50] derivatization. Moreover, we observed a positive correlation between $E_1$ and $17\beta$-$E_2$ across nearly all brain regions examined, as expected. DMIS derivatization has been applied to mouse and human plasma [45,49,72], and human saliva [73], demonstrating its potential applications in other species and sample types.

## Method application in wild songbirds

We applied the assay to wild song sparrows to examine how tissue type, season, and an aggressive encounter affect estrogen levels. In the non-breeding season, estrogens were non-detectable in both CON and STI subjects. In contrast, in the breeding season, both $E_1$ and $17\beta$-$E_2$ were above the LLOQ in the brain in both groups, at higher levels than in circulation, indicating robust tissue-specific differences. Additionally, $E_1$ and $17\beta$-$E_2$ showed pronounced regional variation across the brain, matching known regional differences in aromatase. All of these patterns are consistent with our previous findings in song sparrows under baseline conditions [27,46,50]. However, we found no effect of the aggressive challenge on $E_1$ and $17\beta$-$E_2$ levels in either brain or blood, regardless of season.

Neuroestrogens promote aggression in non-breeding song sparrows, yet we found no evidence of their synthesis during this season. In the song sparrow brain, aromatase is highly expressed year-round [19,28]. Interestingly, in the non-breeding season only, acute aromatase inhibition reduces aggression [29,30] and exogenous $17\beta$-$E_2$ rapidly increases aggressive behavior [31]. Although previous studies suggested that circulating DHEA could serve as an indirect precursor for local estrogen synthesis, our mass spectrometry analyses did not detect DHEA in this species [50]. Instead, other steroids such as progesterone, which we detected in both blood and brain [50], may provide indirect precursors for androgen and estrogen synthesis. Consistent with this idea, we previously reported increases in brain androstenedione and testosterone after 10 min of aggressive interactions using the same behavioral paradigm [27], which could provide the substrate for aromatase to produce neuroestrogens during contests. We therefore predicted that a territorial challenge

would elevate brain estrogens in both seasons. However, this was not observed. In the breeding season, $E_1$ and 17β-$E_2$ were present but unchanged after STI, and in the non-breeding season, brain estrogens were non-detectable in both groups.

One possible explanation is that our sampling time did not coincide with the peak of estrogen production. Subjects were sampled after 10 min of aggressive interaction (plus short playback used to catch the subject), which may have been either too early or too late. Since aromatase is present in the song sparrow brain, its activation may rely on post-translational modifications such as phosphorylation [74,75] to rapidly produce the neuroestrogens. In quail, increased aromatase activity can occur as early as 2 min after aggressive interactions (whose activity also correlates with aggression levels) [10], and is similarly modulated within 1–5 min by sexual interactions, returning to baseline by 15 min [22]. However, in song sparrows, no change in brain 17β-$E_2$ was observed after 5 min of an aggressive challenge under laboratory conditions [76], suggesting that estrogen synthesis may not occur that rapidly in this species. Alternatively, it could be that estrogens take longer times than the 10 min window used here to increase, and we collected tissues too early. In white-crowned sparrows and zebra finches, local 17β-$E_2$ levels and aromatase activity are modulated 30 min after a social interaction [23,25,77]. In song sparrows, studies employing acute pharmacological modulations have used longer timeframes, ranging from 20 min to 24 hr [30,31].

A second possibility is that we did not measure the right analytes. Estrogens are a large and diverse family of compounds, and our method might have omitted critical compounds. For instance, estrogens can be conjugated to glucuronide or sulfate moieties [62]. Neuroestrogens might be synthesized and then promptly converted to these conjugated forms, which we did not measure. It is also possible that freezing and thawing of brain tissue during sample processing caused endogenous estrogens to be metabolized to other analytes, which were not included here [78].

Third, the estrogenic modulation of non-breeding aggression could be due to changes in estrogen sensitivity. Estrogen receptors are expressed in the song sparrow brain during both breeding and non-breeding seasons with no seasonal differences [28]. Interestingly, receptor availability can be modulated by trafficking to and from the cell membrane [79], and receptor function can be rapidly modulated by phosphorylation [80], potentially altering sensitivity to estrogens without requiring changes in ligand concentration. However, this is unlikely due to the changes in behavior observed in aromatase enzyme manipulation experiments.

Lastly, perhaps our stimuli were insufficient to trigger a neuroendocrine response. Subjects were exposed to a live decoy in a transparent cage accompanied by a conspecific song playback. Although this stimulus reliably elicited a robust behavioral aggressive response, it may not have provided the full suite of cues necessary to activate neural estrogen synthesis. For example, studies in quail showed that males exhibit different neuroestrogen patterns depending on whether they have visual access only or full access to females, with copulation producing the strongest effects [81]. Moreover, in our paradigm, the decoy did not directly interact with the subject. In song sparrows, endocrine responses, such as testosterone elevation, depend on the sensory modality of the social cue, with combined visual and auditory stimuli being more effective than either alone [82]. However, previous studies have shown that aromatase inhibition still reduces aggressive responses to this same STI setup [29,30], suggesting that neural estrogens are indeed relevant in this context, even if we were unable to detect them here.

## Conclusions

We developed and validated a highly sensitive LC-MS/MS method for the simultaneous quantification of eleven estrogens in songbird blood and microdissected brain. This approach significantly improves upon previous methods, by correcting for matrix effects and incorporating additional analytes to the panel. Application of the method to free-living male song sparrows exposed to an acute social challenge did not show changes in neuroestrogen levels, despite a strong behavioral response. Our refined method offers a powerful tool for quantifying estrogens in brain, with potential applicability across diverse vertebrate systems.

## Acknowledgments

We thank Asmita Poudel for her assistance with LC-MS/MS, Sofia Gray for help with fieldwork, and Brittany Jensen, Sofia Laforest, and Guillermo Valiño for comments on the manuscript.

## Author contributions

**Conceptualization:** Kiran K. Soma.

**Data curation:** Cecilia Jalabert, Megan Q. Liu.

**Formal analysis:** Cecilia Jalabert, Megan Q. Liu.

**Funding acquisition:** Cecilia Jalabert, Kiran K. Soma.

**Investigation:** Cecilia Jalabert, Megan Q. Liu.

**Methodology:** Cecilia Jalabert.

**Project administration:** Kiran K. Soma.

**Resources:** Kiran K. Soma.

**Supervision:** Kiran K. Soma.

**Validation:** Cecilia Jalabert, Megan Q. Liu.

**Visualization:** Cecilia Jalabert.

**Writing – original draft:** Cecilia Jalabert, Megan Q. Liu.

**Writing – review & editing:** Kiran K. Soma.

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
