## [Decision Letter · Decision Letter 0]

14 Aug 2025

Estrogen profiling in blood and brain: effects of season and an aggressive interaction in a songbird

PLOS ONE

Dear Dr. Soma,

Thank you for submitting your manuscript to PLOS ONE. After careful consideration, we feel that it has merit but does not fully meet PLOS ONE’s publication criteria as it currently stands. Therefore, we invite you to submit a revised version of the manuscript that addresses the points raised during the review process.

We look forward to receiving your revised manuscript.

Kind regards,

Irfan Ahmad Bhat

Academic Editor

PLOS ONE

Journal Requirements:

2. To comply with PLOS One submissions requirements, in your Methods section, please provide additional information regarding the experiments involving birds and ensure you have included details on (1) methods of sacrifice, and (2) efforts to alleviate suffering.

“Agencia Nacional de Investigación e Innovación (POS_EXT_2016_1_134441) to CJ

Zoology Graduate Fellowship, University of British Columbia (6444) to CJ

Canadian Graduate Scholarships-Master's, Canadian Institutes of Health Research (6556) to MQL

Faculty of Medicine Graduate Award, University of British Columbia (6442) to MQL

Canadian Institutes of Health Research Operating Grant (169203) to KKS

Canada Foundation for Innovation Grant (32631) to KKS”

Please state what role the funders took in the study.  If the funders had no role, please state: 'The funders had no role in study design, data collection and analysis, decision to publish, or preparation of the manuscript.'

Additional Editor Comments:

Thank you for submitting your manuscript “Estrogen profiling in blood and brain: effects of season and an aggressive interaction in a songbird.” The study’s methodological contribution in developing a sensitive LC/MS-MS protocol is appreciated, and the application to a songbird model is of interest.

However, both reviewers noted the need for greater clarity in sampling details, breeding status, capture methods, behavioral monitoring, statistical reporting, and equipment specifications. The presentation of results requires improvement, including standardizing units and enhancing figure resolution. Clarification of unexpected findings such as the absence of seasonal or STI-related changes in estrogens is also needed, with reference to previous studies.

Please revise the manuscript to address these points and provide a detailed, point-by-point response.

Reviewers' comments:

Reviewer's Responses to Questions

**Comments to the Author**

1. Is the manuscript technically sound, and do the data support the conclusions?

Reviewer #1: Yes

Reviewer #2: No

2. Has the statistical analysis been performed appropriately and rigorously?

Reviewer #1: Yes

Reviewer #2: No

3. Have the authors made all data underlying the findings in their manuscript fully available?

Reviewer #1: Yes

Reviewer #2: Yes

4. Is the manuscript presented in an intelligible fashion and written in standard English?

Reviewer #1: Yes

Reviewer #2: Yes

Reviewer #1: This paper describes a new highly sensitive method to measure multiple estrogen by aLC/MS_MS. The method appears to be quantitative and precise. It was applied to the song sparrow, a model well-known in the field of Behavioral Neuroendocrinology. In general the paper is very well written and extremely clear. I have no general complaint about the manuscript. Just a few comments about specific items.

I cannot comment in a critical manner about the assay method even if I can understand it but Ihave no personal experience with the different options. This being said, the results are very well documented. Extensive validations have been performed and confirm the accuracy and specificity of the results.

It is said on line 107 that all birds were caught by mist nets and song playback but then on line 218-222, it is said that some birds were caught by STI and other without decoy and loudspeaker./playback . Then a bit later the CON and STI are exposed to playback? How are they different then: Only by the presence/absence of decoy.? Please clarify.

It is unfortunate that estrogens concentrations are expressed in ng/g in figure 3 and in nM in figure 4. This should be standardized one way or the other. If you select the units used in figure 3, I would suggest to change to pg/g since this would remove unnecessary zeros.

One key observation here is that, contrary to expectations, no change in estrogens was observed after 10 min of aggressive interaction while previous studies had observed an increase in testosterone and androstenedione after an STI (line 455), and an increase in E2 had been detected after STI in white-crowned sparrow (line 469). This discrepancy is adequately discussed but It would be important to remind the reader what was the latency in the previous studies compared to this one (see line 462). .

It is also puzzling that all estrogens were undetectable during the non breeding season contrary to what could have been expected based on previous studies of this lab. showing central production of sex steroids from adrenal DHEA. This should also be discussed.

Reviewer #2: The manuscript entitled “Estrogen profiling in blood and brain: effects of season and an aggressive interaction in a songbird" by Jalabert and colleagues established a new protocol for the development of brain estrogens. Subsequently they also attempted to compare brain estrogen levels between breeding and non-breeding season. Topic is more relevant to protocol based journals and biological functions are weakly addressed. My specific comments are:

1. Abstract lacks clarity specifically methodology related with animal experimentation though detailed methodology for establishing new protocol has been provided

2. Line 104-15: It does not clear how many birds were procured at each season.

3. During breeding season samplings were executed for 16 days while non breeding season for a month time (almost double time). What is the rationale behind it? Further, what was the gonadal/reproductive status of the birds procured during the breeding phase? As individual

gonadal status may vary significantly and hence alter the reproduction linked behaviour.

4. What is the natural breeding time in the wild population?

5. Pleas provide protocol number.

6. How individual behaviour of wild bird has been precisely monitored without mixing with

two or more individuals (line 221 to 224).

7. Line 240, please provide details of make and model of cryostat used for sectioning.

8. Statistical analysis section line 256-257 mentioned using one way ANOVA while result

section line 347 to 349 talks about main effect and interaction. These two are contradictory.

9. How individual songs were recorded for wild bird? I don’t see data related to it.

10. Results lacks clarity and graphical resolution and representation is poor. It lacks

necessary information.

11. Despite robust aggressive responses to the STI, estrogen levels did not differ between

STI and control subjects in either season. Could it be because of sensitivity of the assay?

**Do you want your identity to be public for this peer review?** For information about this choice, including consent withdrawal, please see our Privacy Policy

Reviewer #1: No

Reviewer #2: No

---

## [Author Response · Author response to Decision Letter 1]

8 Oct 2025

Please see attached rebuttal letter.

---

## [Decision Letter · Decision Letter 1]

3 Nov 2025

Estrogen profiling in blood and brain: effects of season and an aggressive interaction in a songbird

PONE-D-25-30431R1

Dear Dr. Soma,

We’re pleased to inform you that your manuscript has been judged scientifically suitable for publication and will be formally accepted for publication once it meets all outstanding technical requirements.

Kind regards,

Irfan Ahmad Bhat

Academic Editor

PLOS ONE

Additional Editor Comments (optional):

The reviewers has suggested to accept the manuscript for publication but as per reviewer 2 figures 3 and 4 needs some refinement that can be done during the proof submission.

Reviewers' comments:

Reviewer's Responses to Questions

**Comments to the Author**

Reviewer #1: All comments have been addressed

Reviewer #2: All comments have been addressed

2. Is the manuscript technically sound, and do the data support the conclusions?

Reviewer #1: Yes

Reviewer #2: Partly

3. Has the statistical analysis been performed appropriately and rigorously?

Reviewer #1: Yes

Reviewer #2: Yes

4. Have the authors made all data underlying the findings in their manuscript fully available?

Reviewer #1: Yes

Reviewer #2: Yes

5. Is the manuscript presented in an intelligible fashion and written in standard English?

Reviewer #1: (No Response)

Reviewer #2: Yes

Reviewer #1: The authors did a very good job in answering all comments that had been presented by the two reviewers. The changes made in the manuscript are in my opinion answering all questions that had been presented. For me this paper could be accepted and published in its present form. I have, however, two remaining comments for the authors.

a) I understand the rationale in presenting concentrations in nM if you want to compare different steroids even if in this case the differences of molecular weight of the different estrogens are minimal. But then why keep figure 3 in ng/g since like figure 4 it is comparing E2 and E1. It might be simpler to have all figures in pg or pg/g or in nM pr nanomoles(for fig 1). The numbers in figure 3 would be simpler if expressed in pg/g instead on ng/g, this would remove all leading zeros.

b) The discussion starting on line 481 of the fact that sampling might have missed a critical increase adequately covers the question and now I fully understand the procedure. However, I am still wondering whether an increase in one or more estrogens could not be induced by the short playback used to capture birds. This is not the case in laboratory conditions but may be this could occur in the wild when birds are often more responsive? If this was the case then the peak of estrogens might induced in the same manner by the capture procedure thus masking any effect of the longer STI? If that was the case, then there should be a positive correlation between estrogen concentrations and the latency to catch the bird after the beginning of the playback I guess the fact that this latency did not differ between the STI and control birds pleads against such an interpretation. . Does this make sense?

These are however only suggestions and I would hate that request another revision for such minimal remarks. Authors may want to deal with these suggestions by making minor additions at the proof stage (change figure(s) for point a and add a couple of sentence for point b). Or may be it would be sufficient to have these comments published with the review history of the paper.

Reviewer #2: Thank you for addressing my previous comments. I have minor suggestions for resolution and pattern used for figure 3 and 4. It is difficult to see individual data points in current format and authors may select a different pattern for Figure 3 and may be hollow and solid circles for figure 4.

---

## [Editor Report · Acceptance letter]

PONE-D-25-30431R1

PLOS ONE

Dear Dr. Soma,

I'm pleased to inform you that your manuscript has been deemed suitable for publication in PLOS ONE. Congratulations! Your manuscript is now being handed over to our production team.

Kind regards,

on behalf of

Dr. Irfan Ahmad Bhat

Academic Editor

PLOS ONE